# Characterization of the Monkeypox Virus [MPX]-Specific Immune Response in MPX-Cured Individuals Using Whole Blood to Monitor Memory Response

**DOI:** 10.3390/vaccines12090964

**Published:** 2024-08-26

**Authors:** Elisa Petruccioli, Settimia Sbarra, Serena Vita, Andrea Salmi, Gilda Cuzzi, Patrizia De Marco, Giulia Matusali, Assunta Navarra, Luca Pierelli, Alba Grifoni, Alessandro Sette, Fabrizio Maggi, Emanuele Nicastri, Delia Goletti

**Affiliations:** 1Translational Research Unit, National Institute for Infectious Diseases “Lazzaro Spallanzani” IRCCS, 00149 Rome, Italy; elisa.petruccioli@inmi.it (E.P.); settimia.sbarra@inmi.it (S.S.);; 2Highly Infectious Diseases Isolation Unit, Clinical Department, National Institute for Infectious Diseases “Lazzaro Spallanzani” IRCCS, 00149 Rome, Italy; 3Laboratory of Virology and Biosafety Laboratories, National Institute for Infectious Diseases “Lazzaro Spallanzani” IRCCS, 00149 Rome, Italy; 4Clinical Epidemiology Unit, National Institute for Infectious Diseases “Lazzaro Spallanzani” IRCCS, 00149 Rome, Italy; 5Unità Operativa Complessa (UOC) Transfusion Medicine and Stem Cell, San Camillo Forlanini Hospital, 00149 Rome, Italy; 6Center for Vaccine Innovation, La Jolla Institute for Immunology, La Jolla, CA 92037, USA; 7Department of Medicine, Division of Infectious Diseases and Global Public Health, University of California, San Diego (UCSD), La Jolla, CA 92037, USA

**Keywords:** Mpox, MPXV, IFN-γ

## Abstract

Background: Monkeypox (Mpox) is a zoonotic disease caused by monkeypox virus (MPXV), an Orthopoxvirus (OPXV). Since we are observing the first MPXV outbreak outside the African continent, the general population probably does not have a pre-existing memory response for MPXV but may have immunity against the previous smallpox vaccine based on a live replicating Vaccinia strain (VACV). Using a whole blood platform, we aim to study the MPXV- T-cell-specific response in Mpox-cured subjects. Methods: We enrolled 16 subjects diagnosed with Mpox in the previous 3–7 months and 15 healthy donors (HD) with no recent vaccination history. Whole blood was stimulated overnight with MPXV and VACV peptides to elicit CD4 and CD8 T-cell-specific responses, which were evaluated by ELISA and multiplex assay. Results: Mpox-cured subjects showed a significant IFN-γ T-cell response to MPXV and VACV. Besides IFN-γ, IL-6, IP-10, IL-8, IL-2, G-CSF, MCP-1, MIP1-α, MIP-1β, IL-1Rα, and IL-5 were significantly induced after specific stimulation compared to the unstimulated control. The specific response was mainly induced by the CD4 peptides MPX-CD4-E and VACV-CD4. Conclusions: We showed that MPXV-specific responses have a mixed Th1- and Th2-response in a whole blood platform assay, which may be useful for monitoring the specific immunity induced by vaccination or infection.

## 1. Introduction

Monkeypox virus (MPXV) is an Orthopoxvirus (OPXV) belonging to the Poxviridae family. Symptoms include fever, myalgia, lymphadenopathies, and skin and mucosal rash similar to those caused by infection with the Variola virus (VARV), the etiological agent of smallpox [1,2,3]. The World Health Organization recently declared Mpox disease a public health emergency of international concern [4].

Since the 2022 global outbreak, more than 97,208 people have been infected with MPXV, with a high proportion of cases reported from countries without previously documented MPXV transmission, such as areas of West or Central Africa [1]. Except for West and Central African countries, the ongoing outbreak of Mpox mainly affects men who have sex with men (MSM) [1].

Currently, Mpox disease incidence has decreased by 90% compared with the outbreak’s peak numbers in 2022; however, sustained community transmission continues mainly in Southeast, Asia and the Western Pacific regions [1].

In 2022, we observed the first MPXV outbreak outside the African continent [5,6], and it is therefore likely that the general population did not have a pre-existing memory immune response specific to MPXV. However, the worldwide population was vaccinated against smallpox until the 1970s, with a closely related OPXV, the vaccinia virus (VACV). VACV stimulates the immune response, inducing cross-protective immunity against smallpox and other OPXVs, including MPXV [7,8,9,10,11]. Individuals unvaccinated against smallpox but probably exposed to other poxviruses, such as molluscum contagiosum virus (MCV), exhibit an in vitro response to MPXV-specific stimulation [9,10,12], suggesting a potential role for T cell cross-reactivity in controlling a new pox infection.

It has been demonstrated that both humoral and cellular responses are important for controlling OPXV infections [13,14,15,16,17,18,19]. Considering that MPXV re-infection is unusual [20], developing a memory immune response may help clear a new potential infection. Evidence of a long-term memory immune response, even decades after smallpox vaccination, supports this hypothesis [9,10,21]. Moreover, a recent study on Mpox convalescent patients demonstrated that MPXV infection induces a T-cell response with a polyfunctional and cytotoxic T-cell response with an effector memory phenotype [9]. Whole blood tests to detect T-cell responses are largely used in diagnostics and research [22]. Currently, no studies on MPXV have been done using this platform.

Therefore, to better understand long-term MPXV immunity, we studied the T-cell response to MPXV in convalescent Mpox subjects using a whole blood platform with targeted MPXV peptide pools, and compared these with VACV peptide pools [12].

## 2. Materials and Methods

### 2.1. Standard Protocol Approvals, Registrations, and Patient Consents

The Lazzaro Spallanzani National Institute for Infectious Diseases IRCCS (INMI) Ethical Committee approved the human study protocols (43z del Registro delle Sperimentazioni Non-COVID 2022 amended with approval 11z del Registro delle Sperimentazioni Non-COVID 2023). The study protocols were written following the ethical principles for human experimentation according to the Declaration of Helsinki. All participants signed written informed consent.

### 2.2. Study Population

Thirty-one individuals were prospectively recruited for the study. Among the healthy donors, thirteen were healthcare workers at INMI with no exposure to Mpox patients, and two individuals were blood donors from San Camillo Hospital in Rome. Sixteen individuals with Mpox disease in the previous 3–7 months, followed at the outpatient clinic of INMI, were enrolled as Mpox-cured individuals between November 2022 and April 2023.

### 2.3. Definition of MPX Disease

Mpox diagnosis was based on the presence of Mpox DNA by PCR testing in clinical specimens.

### 2.4. Peptide Pools for the Whole Blood Assay

The MPXV and Vaccinia (VACV) peptide mega pools (MPs) were provided by the laboratory of Alessandro Sette [La Jolla Institute for Immunology (LJI), La Jolla, CA, USA] [12]. OPXV-CD4-E and OPXV-CD8-E pools were based on experimentally defined Orthopox T cell epitopes mostly derived from Vaccinia virusù and were renamed in the current manuscript as VACV-CD4 and VACV-CD8 pools. The MPX-CD4-P and MPX-CD8-P peptide pools were predicted T cell epitopes from the most immunodominant VACV ortholog proteins for MPXV [12]. CD4 peptide pools encompass 15-mer peptides able to elicit a CD4 T-cell response but can also induce a CD8 T-cell response. Conversely, CD8 peptide pools were 9- and 10-mers, thus eliciting a specific CD8 T-cell response. MPX-CD8-P1, MPX-CD8-P2, MPX-CD8-P3, MPX- CD8-P4, and MPX-CD8-P5 were pooled together (MPX-CD8-P) to measure the MPXV-specific CD8 T-cell response. The peptide pools were used at 1 ug/mL.

### 2.5. IFN-γ Whole Blood Assay

To measure IFN-γ production after antigen-specific stimulation, whole blood (500 μL) was incubated in a 48-well flat-bottom plate with different peptide pools of OPXV. As an experimental positive control, we used staphylococcal enterotoxin B (SEB) (Sigma-Aldrich, Milan, Italy) (200 ng/mL). After overnight (20–24 h) incubation at 37 °C (5% CO_2_), the plasma was collected and frozen at −80 °C. IFN-γ levels were measured by enzyme-linked immunosorbent assay (ELISA) using the QuantiFERON (QFT)-Plus assay (Diasorin, Vercelli, Italy), following the manufacturer’s instructions. The detection limits of the assay were 0.065 and 10 IU/mL.

### 2.6. Humoral Response

Immunoglobulin (Ig) titers were evaluated by performing an indirect immunofluorescence assay (IFA) (secondary antibodies FITC conjugated from Euroimmun) for anti-MPXV IgM, IgA, and IgG on homemade slides containing MPXV-infected VeroE6 cells [23]. Neutralizing antibodies (Nabs) against MPXV were evaluated in subjects with a positive IgG titer, as described [23].

### 2.7. Multiplex Analysis

We selected a subgroup of 10 Mpox-cured individuals to perform a multiplex assay. Based on the available samples, we selected 5 subjects living with HIV and 5 HIV-uninfected subjects. A multiplex immune assay was performed using plasma harvested from the different experimental conditions. The Bio-Plex Pro Human Cytokine 27-plex Assay panel and the MagPix system (Bio-Rad, Hercules, CA, USA) were used to measure the levels of several immune factors: [IL-1RA, IL-1β, IL-4, IL-2, IL-5, IL-7, IL-6, IL-9, IL-10, IL-8, IL-12p70, IL-17A, IL-15, IL-13, eotaxin, granulocyte-colony stimulating factor (G-CSF), basic fibroblast growth factor (FGF), IFN-γ, IP-10, macrophage inflammatory protein (MIP)-1α, monocyte chemoattractant protein-1 (MCP-1), MIP-1β, granulocyte-macrophage colony-stimulating factor (GM-CSF), vascular endothelial growth factor (VEGF), platelet-derived growth factor (PDGF), RANTES (regulated on activation, normal T cell expressed and secreted), and tumor necrosis factor-alpha (TNF)-α]. Raw data were generated using Bio-Plex Manager software, version 6.2. Values below the detection range were considered zero, and values above the detection range were changed to the highest value of the standard curve. Values from the unstimulated negative control were subtracted from each condition. All samples with an acquired bead count of less than 50 were excluded from the analysis.

### 2.8. Statistical Analysis

Data were analyzed using Graph Pad (GraphPad Prism 8 XML Project). The median and interquartile ranges (IQRs) were calculated for continuous measures; categorical variables were reported as count and proportion. Non-parametric statistical inference tests were applied. In case of unmatched comparison among several groups, Kruskal–Wallis and Mann-Whitney tests were used. For matched comparison, the Wilcoxon matched-pairs signed rank test was used. The Chi-Square and Fisher tests were used for proportions.

## 3. Results

### 3.1. Clinical and Epidemiological Characteristics of the Study Population

We enrolled 31 subjects (Table 1): 15 HD and 16 subjects with a past Mpox disease in the previous 3–7 months, defined as “Mpox-cured individuals”. No significant differences were found in terms of age and origin; however, the HD group mostly included women, whereas the cured Mpox group included only males (*p* < 0.0001). HD subjects were selected to be VACV-unvaccinated, whereas one Mpox-cured subject was VACV-vaccinated. To characterize the immune response to MPXV, we also evaluated the serology response: most of the Mpox individuals showed a positive IgG and Nabs response and 7 individuals had a positive IgA response. The only subject with a positive IgM response had a negative IgA response and a positive IgG response. As expected, no response was found among the HD. Among the Mpox-cured subjects, seven were people living with HIV infection (PLWH) with a median CD4 cell count of 754 CD4/mm^3^ and less than 30 copies/mL of HIV-RNA in the plasma.

### 3.2. MPXV-Specific T-Cell Response in Mpox-Cured Individuals

To assess the ability of Mpox-cured individuals to respond to MPXV, we evaluated the IFN-γ response differently induced by the MPXV peptide pools available, compared with the unstimulated conditions (Figure 1).

Mpox-cured subjects showed significant IFN-γ production compared to the unstimulated condition in response to MPX-CD4-P (median 0.160, IQR 0.08250–0.5100, *p* < 0.0002); MPX-CD8-P (median 0.1000, IQR 0.06000–0.2300, *p* < 0.0354), VACV-CD4 (median 0.1700, IQR 0.1125–0.6275, *p* < 0.0001) and VACV-CD8 (median 0.09000, IQR 0.07000–0.2075, *p* = 0.0103) (Figure 1).

Among the MPX-CD8 peptide pools, MPX-CD8-P3 and MPX-CD8-P4 were the most immunogenic CD8 peptide-pools compared to the others (Appendix A). HIV infection did not affect the MPXV T-cell response in terms of the magnitude of the response (Appendix A).

To verify the humoral response of enrolled Mpox-cured individuals, we evaluated the IgM, IgA, IgG, and Nabs responses between PLWH and HIV-uninfected subjects (Appendix A). Interestingly, we found a similar antibody response between PLWH and HIV-uninfected subjects, reflecting the similar MPXV T-cell response detected.

### 3.3. Multiplex Analysis of Immune Factors Different from IFN-γ Specifically Induced after In Vitro Mpox Peptide Stimulation in Mpox-Cured Subjects

In a subgroup of 10 Mpox-cured subjects, we evaluated the plasma levels of 27 cytokines, chemokines, and growth factors after specific stimulation.

We found a significant induction of IL-6, IFN-γ, IP-10, IL-8, IL-2, G-CSF, MCP-1, MIP1-a, MIP-1b, IL-1Rα, and IL-5 in response to MPXV peptide stimulation compared with the unstimulated control (Figure 2). No significant modulations were found for the other factors. Analyzing the proportion of response to each stimulation, the antigen-specific response was mainly observed in CD4 peptide pools (Figure 3, Appendix A).

IL-2 levels showed the highest percentage change in response to MPX-CD4-P and VACV-CD4 compared to the unstimulated conditions, whereas the modulation of IL-6, IL-2, IL-8, and MCP-1 characterized the response to MPX-CD8-P and VACV-CD8 (Appendix A).

## 4. Discussion

The induction of a functional antigen-specific T-cell response is crucial for clearing viral infections and developing memory immunity. Few studies are available on the T-cell immunity associated with MPXV natural infection. In this study, we showed that the MPXV-specific response can be measured using a whole blood platform with MPXV and VACV peptides designed to elicit CD4 and CD8 T-cell responses. The response was mainly mediated by CD4-specific T-cells. HIV infection did not affect the response. These results may contribute to designing standardized tests to detect immune responses to MPXV.

Evidence generated in animal models demonstrated the importance of CD8 T-cells in MPXV clearance [24], whereas a study performed on Mpox patients highlighted the presence of a memory polyfunctional CD4 T-cell response to MPXV-specific peptide stimulation [9]. Here we show a higher magnitude of IFN-γ production and a higher number of responders to CD4-specific peptide stimulation (MPX-CD4-P and VACV-CD4 peptides) compared to CD8-specific peptide stimulation (MPX-CD8-P and VACV-CD8 peptides). The multiplex immune factor evaluation supported these findings, showing a major involvement of CD4 peptide-specific response in the induction/modulation of an immune response characterized by the release of several chemokines, cytokines, and growth factors. Indirectly, these data suggest a higher contribution of the antigen-specific CD4 T-cell response in Mpox-cured individuals, as shown in other infectious diseases [25,26]. These data are in agreement with recent cytometry studies in Mpox convalescent individuals, which showed the engagement of CD4 T-cell response to the same MPXV peptides used in this study [7,9].

The serology response of Mpox-cured individuals was characterized by the presence of IgG and Nabs and the absence of an IgM response, except in one subject. Moreover, 43% of the cured Mpox subjects still showed a positive IgA response. These results are supported by recent findings from our institution, showing that in Mpox individuals, 6-8 months after symptom onset, the IgM response became undetectable, the IgA response was detected in 25% of subjects, and IgG and nAb decreased but remained positive in all individuals [27,28]. We evaluated the Ig titers performing an IFA on homemade slides containing MPXV-infected VeroE6 cells, and, likely this led to a more sensitive assay compared to a previous study reporting an important decline in both IgG and IgA responses after 6 months [29].

In this experimental setting, HIV infection status did not affect the MPXV-mediated response in terms of INF-γ production and antibody response, likely due to the good CD4 T-cell counts and undetectable HIV load. These results are supported by recent findings showing that the T-cell response against MPXV was detectable up to 6 months after diagnosis, independently of HIV status [29]. Although this result needs to be confirmed in larger cohorts and the enrolled PLWH were not immunosuppressed, we think it is important to verify the immunity to MPXV in this cohort of patients.

Innate immune cells typically represent the first defense against viruses; however, these cells are also targets for MPXV, facilitating virus dissemination in the host. Consequently, the detection of MPXV antigens in innate cells such as monocytes and neutrophils has been associated with MPXV lethality [30,31,32,33]. The cells intended to fight the infection act as Trojan horses and induce the immune-specific response, including the release of several chemokines and cytokines that amplify the response and recruit more neutrophils and monocytes at the site of infection. The increased pro-inflammatory milieu leads to T-cell differentiation toward a Th2 and Th1 profile. The Th2 response balances the Th1 response induced by the virus infection, as shown by the increased Th2 cytokine response and reduced Th1 response [34]. Moreover, it has been demonstrated that MPXV can inhibit T-cell activation and the production of pro-inflammatory cytokines [35]. Therefore, several mechanisms contribute to the pathogenesis and spreading of infected cells.

After the interaction with antigen-presenting cells, naïve T-cells may differentiate toward T-helper 1 (Th1) or T-helper 2 (Th2) cells [36,37]. Th1 cells produce IL-2, TNF-α, and IFN-γ, whereas IL-4, IL-5, and IL-10 are considered Th2 cytokines [36,38]. IL-6 is an inflammatory cytokine produced by several antigen-presenting cells, acting as a fundamental regulator of Th1-Th2 differentiation [39]. Th1-type cytokines control proinflammatory responses and activate intracellular killing mechanisms, whereas Th2 cytokines facilitate IgG1 and IgE antibody production and eosinophilic responses. Th2 responses have several functions, including the containment of an excessive Th1 response.

Through multiplex analysis, we demonstrated significant modulation of several cytokines, chemokines, and growth factors involved in the Th1 network, such as IFN-γ, IL-2, and IP-10 [40], or those mainly secreted by activated macrophages, such as MIP-1a and MIP-1b [41]. Among the pro-inflammatory cytokines, IL-8, which is induced by IL-1 and TNF-α [42], plays a role in neutrophil activation and is involved in the pathogenesis and progression of hyperinflammation and acute respiratory syndrome [43]. The effect of pro-inflammatory cytokines is balanced by IL-1Ra activity, which is upregulated by proinflammatory mediators such as IL-1 itself and induces strong anti-inflammatory effects [44]. The high levels of G-CSF and IL-5 specific responses to MPXV peptide stimulation are in agreement with the elevated number of monocytes and granulocytes [eosinophils, basophils, and neutrophils] in the blood of patients with severe monkeypox disease [45,46]. Elevated levels of MCP-1 supported a Th2 polarization [47], as demonstrated by the presence of IL-5 [48]. Studies in animal models have shown that severe MPXV infection induces a sustained cytokine storm characterized by an overproduction of pro-inflammatory mediators and cytokines [49]. A cytokine storm is defined as a dysregulated immune response characterized by systemic inflammation status, specific symptoms, and eventually multiorgan failure [50]. In severe Mpox disease, the cytokine storm is mainly characterized by a Th2 profile, with high serum levels of IL-4, IL-6, IL-5, and IL-10 and a reduction of Th1 cytokines [34,46,51]. In this context, the release of IL-10 balances the inflammatory response by suppressing the activity of neutrophils [51]. Although none of the cured Mpox individuals enrolled in our cohort had severe symptoms, we reported significant modulation of IL-5, indicating the development of a Th2 memory response to fight the virus infection.

Our multiplex data describe a Th1 and Th2 profile, showing the ability of immunity to balance an excessive Th1 response. IL-2 is highly induced after MPXV-specific stimulation and as known, is a key cytokine in the Th1 network.

Thus, IL-2, either alone or in combination with IFN-γ, may be considered a potential immune biomarker to study the MPXV-specific response over time.

Considering that we enrolled only cured Mpox individuals, we could not compare the memory immunity of convalescent individuals with the immune response during the acute phase of Mpox disease. Despite this limitation and the low number of cured Mpox individuals enrolled, we contributed to building a picture of MPXV immunity, demonstrating that MPXV-CD4 peptides induce a Th1 and Th2 response in a whole blood platform assay. Based on this evidence, we propose an easy tool to study the MPXV-specific CD4 and CD8 T-cell response, potentially useful for monitoring MPXV immunity in vaccine trials.

## Figures and Tables

**Figure 1 vaccines-12-00964-f001:**
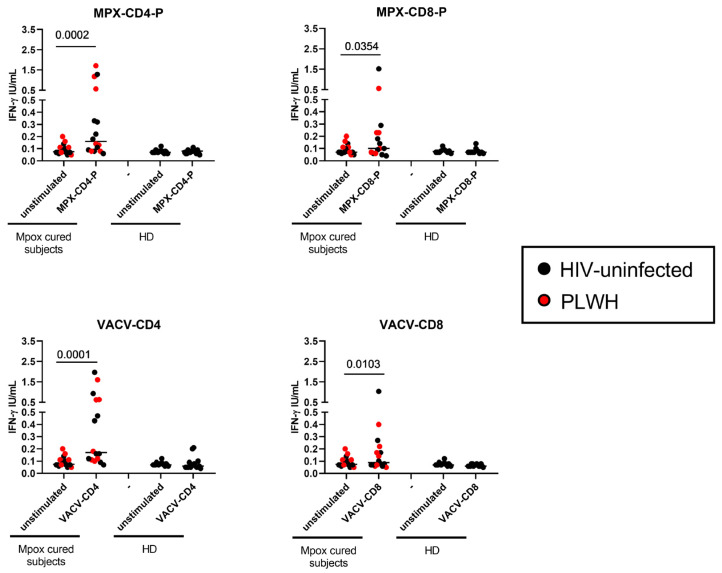
MPXV-specific T-cell response from in vitro stimulated samples from Mpox-cured individuals. Graphs report the IFN-γ response induced by MPX-CD4-P; MPX-CD8-P, VACV-CD4, and VACV-CD8 stimulation. Results for the MPX-CD8-P condition were available only for 15 individuals. ELISA was performed on plasma samples from whole blood stimulation, and IFN-γ was expressed as IU/mL; the IFN-γ values of the stimulated conditions were not subtracted from the unstimulated control values. The horizontal lines represent the median; statistical analysis was performed using the Wilcoxon test; black plots refer to HIV-uninfected subjects, and red plots refer to PLWH. **Abbreviations:** IFN-γ: interferon-γ; HD: healthy donor; PLWH: people living with HIV.

**Figure 2 vaccines-12-00964-f002:**
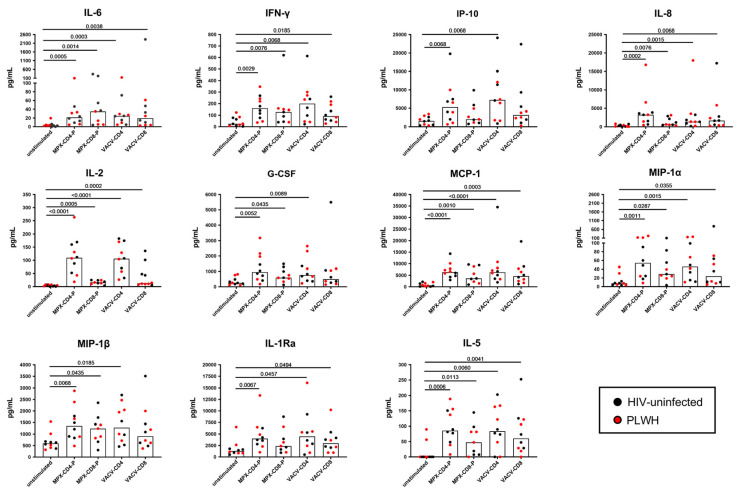
Multiplex analysis of immune factors different from IFN-γ specifically induced after in vitro Mpox peptide stimulation in Mpox-cured subjects. Graphs report the levels of different immune factors in response to MPX-CD4-P; MPX-CD8-P, VACV-CD4, and VACV-CD8 stimulation. The different immune factors were evaluated by Luminex assay in plasma collected after whole blood stimulation, and levels of analytes were expressed as IU/mL. The value of the stimulated condition was not subtracted from the value of the unstimulated control. Among the 27 analytes evaluated, we report only the immune factors with significant modulation between the unstimulated control and peptide stimulation. The horizontal lines represent the median; statistical analysis was performed using the Wilcoxon test; black plots refer to HIV-uninfected subjects, and red plots refer to PLWH. **Abbreviations:** IFN-γ: interferon-γ; IL: interleukin; IP-10: interferon-γ inducible protein; G-CSF: granulocyte-colony-stimulating factor; MCP-1: monocyte chemoattractant protein-1; MIP-1: monocyte chemoattractant protein-1; IL-1Ra: interleukin-1 Receptor antagonist; PLWH: people living with HIV.

**Figure 3 vaccines-12-00964-f003:**
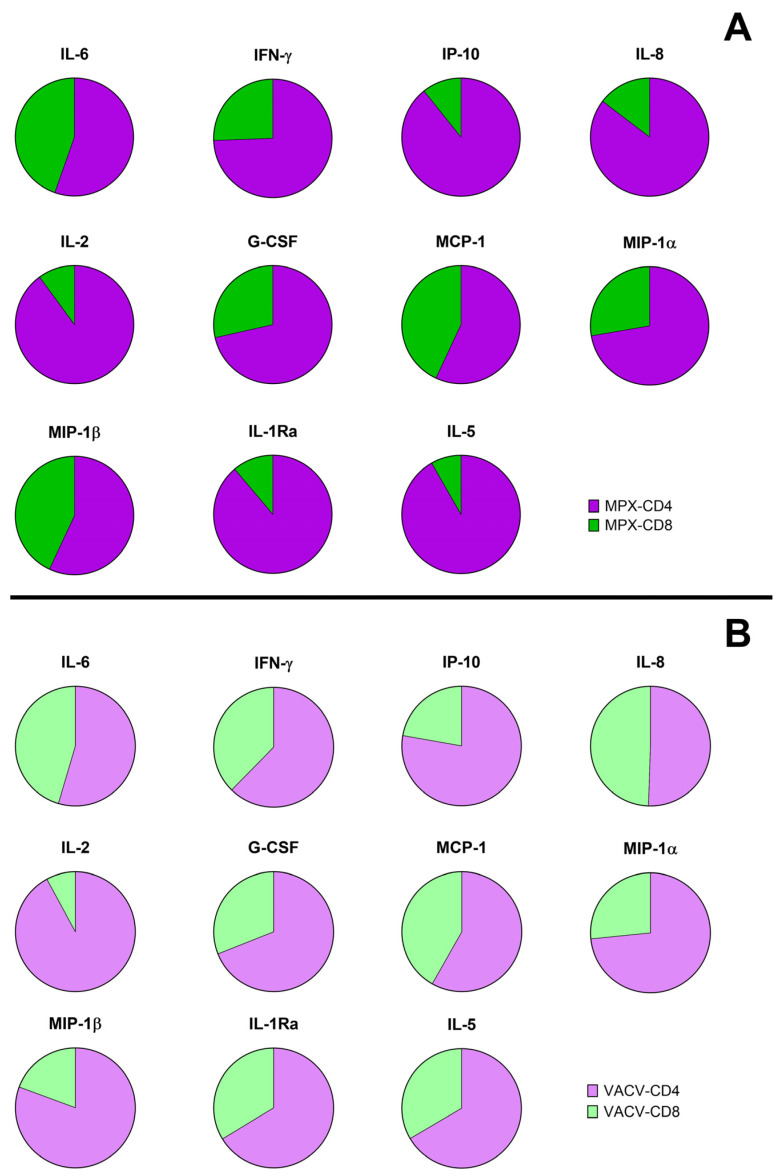
Mpox-specific immune signature based on selected immune factors. (**A**) The graphs represent the median proportion of immune factors secreted in response to (**A**) MPX-CD4-P; MPX-CD8-P; (**B**) VACV-CD4, VACV-CD8. The different immune factors were measured by Luminex assay in plasma collected after whole blood stimulation. The value of the stimulated condition was subtracted from the value of the unstimulated control. **Abbreviations:** IFN-γ: interferon-γ; IL: interleukin; IP-10: interferon-γ inducible protein; G-CSF: granulocyte-colony stimulating factor; MCP-1: monocyte chemoattractant protein-1; MIP-1: macrophage inflammatory protein-1; IL-1Ra: interleukin-1 Receptor antagonist.

**Table 1 vaccines-12-00964-t001:** Clinical and epidemiological characteristics of enrolled subjects.

	Healthy Controls	Cured-Mpox Subjects	Total	*p*
**Number (%)**	15	16	31	
**Median age (IQR)**	36 (30–42)	39.5 (36–48)	38 (32–44)	0.1163 *
**Male gender (%)**	4 (27)	16 (100)	20 (64)	<0.0001 **
**Origin N (%)**				0.3671 **
**Western Europe**	14 (93)	15 (94)	29 (94)	
**Eastern Europe**	0 (0)	1 (6)	1 (3)	
**Middle East**	1 (7)	0 (0)	1 (3)	
**Vaccinated with Vaccinia virus (%)**	0 (0)	1 (6.25)	1 (3.2)	na
**MPXV-Serology** **N of reactive (%)**				
**IgM**	nav	1 (6.25)		nav
**IgA**	nav	7 (43.75)		nav
**IgG**	0 (0)	16 (100)		nav
**Nabs**	nav	^§^ 14 (93)		nav
**HIV-infection (%)**	0 (0)	7 (44)	7 (23)	nav
**CD4 count/mm^3^ median (IQR)**	/	754 (711–1000)	/	nav
**HIV-RNA log10 copies/mL (plasma)**	/	<30	/	nav

MPX: monkeypox; na: not applicable for inclusion criteria of the HD group; nav: not available; Ig: immunoglobulin; Nabs: Neutralizing antibodies. * Mann Whitney test ** Fisher exact test. ^§^ Nabs percentage evaluated in 15 total subjects with a positive MPXV-IgG; data not available for 1 subject.

## Data Availability

The raw data generated and/or analysed in the present study are available in our institutional repository (rawdata.inmi.it), subject to registration. The data can be found by selecting the article of interest from a list of articles ordered by the year of publication. No charge for granting access to data is required. In the event of a malfunction of the application, the request can be sent directly by e-mail to the library (biblioteca@inmi.it).

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
