# Peer review of "Characterization of the Monkeypox Virus [MPX]-Specific Immune Response in MPX-Cured Individuals Using Whole Blood to Monitor Memory Response"

_vaccines, 2024, doi:10.3390/vaccines12090964_

Round 1

Reviewer 1 Report

Comments and Suggestions for Authors

This is an interesting research describing the “Characterization of the monkeypox virus [MPX]-specific immune response in MPX-cured individuals in whole blood to monitor memory response” The overall manuscript is a valuable addition to the field with minor revisions to address the mentioned weaknesses, will significantly contribute to our understanding of Monkeypox virus. I recommend considering it for publication after addressing these key points.

Major:

>The cytokines production response to MPXV were all elevated, the author took the whole blood (plasma) which contain peripheral blood mononuclear cells (PBMCs) (i.e. lymphocytes, monocytes, natural killer cells (NK cells) or dendritic cells). The MPXV-specific responses showed mixed Th1- and Th2-response, which cell subset is involved in high level induction of cytokines and chemokines response to MPXV?

>The author should specify the experimental replication for ELIZA test?

Minor:

>The manuscript needs grammatical modifications specially punctuations.

>Line 45, combine the numeric words. 90 439.

>Check the space b/w words throughout the manuscript, such as line 60, 78, 96, 98, 102, 206 etc.

>Line 166, what is the level of significance in cytokines and chemokines test (P<0.0001 or P =0.0001)?

Comments on the Quality of English Language

I recommend considering this manuscript for publication after addressing some key points mentioned in comments and suggestions for author section. thanks

Author Response

Comments and Suggestions for Authors

This is an interesting research describing the “Characterization of the monkeypox virus [MPX]-specific immune response in MPX-cured individuals in whole blood to monitor memory response” The overall manuscript is a valuable addition to the field with minor revisions to address the mentioned weaknesses, will significantly contribute to our understanding of Monkeypox virus. I recommend considering it for publication after addressing these key points.

 Major:

>The cytokines production response to MPXV were all elevated, the author took the whole blood (plasma) which contain peripheral blood mononuclear cells (PBMCs) (i.e. lymphocytes, monocytes, natural killer cells (NK cells) or dendritic cells). The MPXV-specific responses showed mixed Th1- and Th2-response, which cell subset is involved in high level induction of cytokines and chemokines response to MPXV?

Thank you for the comment. Innate immune cells typically represent the first defense against viruses; however, these cells are also a target for MPXV, facilitating the virus dissemination in the host (Lum et al, Nature 2022). Consequently, the detection of MPXV antigens in innate cells such as monocytes and neutrophils has been associated with MPXV lethality (Lum, Nature 2022). The cells intended to fight the infection act as Trojan horses and induce the immune-specific response, including the release of several chemokines and cytokines that amplify the response and recruit more neutrophils and monocytes at the site of infection. The increased pro-inflammatory milieu leads to a T-cell differentiation toward a Th2 and Th1 profile. Th2 response balances the Th1 response induced by the virus infection, as shown by the increased Th2 cytokine response and the reduced Th1 response (Johnston et al, Journal of Clinical Virology 205).   We add a comment in the discussion.

The author should specify the experimental replication for ELIZA test?

Thank you for the comment, due to the low blood sample volume, the ELISA experiments were done using single samples without including replicating conditions. However, since the kit to perform the ELISA is a commercial standardized procedure, we think that the results are reliable.

Minor:

>The manuscript needs grammatical modifications specially punctuations.

>Line 45, combine the numeric words. 90 439.

Thank you for the comment, we modified the text accordingly.

>Check the space b/w words throughout the manuscript, such as line 60, 78, 96, 98, 102, 206 etc.

Thank you for the comment, we modified the text accordingly.

>Line 166, what is the level of significance in cytokines and chemokines test (P<0.0001 or P =0.0001)?

Thank you for the comment. If the referee means the level of IFN-γ produced following MPXV- and VACV-peptide stimulation, we added this information in the text; the statistical significance for each stimulus is already stated in the text and the graphs (Figure 1).

Comments on the Quality of English Language

I recommend considering this manuscript for publication after addressing some key points mentioned in comments and suggestions for author section. thanks

Reviewer 2 Report

Comments and Suggestions for Authors

The manuscript titled " Characterization of the monkeypox virus [MPX]-specific immune response in MPX-cured individuals in whole blood to monitor memory response " addresses the T-cell-specific immune response in individuals who have recovered from monkeypox (Mpox) by using a whole-blood platform. The results showed that MPXV-specific responses have a mixed Th1- and Th2-response in a whole-blood platform assay, which can be useful to monitor the specific immunity induced by vaccination and infection. The manuscript is well-structured, providing clear descriptions of the methodology, results, and interpretation of the findings. I believe this manuscript is publishable after minor revisions, such as increasing the sample size if possible and including a control group with recent smallpox vaccinations.

1. The sample size is relatively small (16 Mpox subjects and 15 healthy donors), which might limit the generalizability of the findings. Increasing the sample size could strengthen the statistical power of the study.

2. The study includes healthy donors without a recent vaccination history, but it would be beneficial to include a control group with recent smallpox vaccinations for comparison.

Author Response

REVIEWER 2

Comments and Suggestions for Authors

The manuscript titled " Characterization of the monkeypox virus [MPX]-specific immune response in MPX-cured individuals in whole blood to monitor memory response " addresses the T-cell-specific immune response in individuals who have recovered from monkeypox (Mpox) by using a whole-blood platform. The results showed that MPXV-specific responses have a mixed Th1- and Th2-response in a whole-blood platform assay, which can be useful to monitor the specific immunity induced by vaccination and infection. The manuscript is well-structured, providing clear descriptions of the methodology, results, and interpretation of the findings. I believe this manuscript is publishable after minor revisions, such as increasing the sample size if possible and including a control group with recent smallpox vaccinations.

  1. The sample size is relatively small (16 Mpox subjects and 15 healthy donors), which might limit the generalizability of the findings. Increasing the sample size could strengthen the statistical power of the study.

Thank you for the comment. We are aware of this limit. The ongoing MPXV epidemic does not include Italy; therefore, we cannot enroll new patients. Although the study has been performed in a small cohort of MPX-cured individuals, we contributed to building a picture of MPXV immunity, demonstrating that MPXV-CD4 peptides induce a Th1 and Th2 response in a whole-blood platform assay. We highlighted this limit in the discussion.

  1. The study includes healthy donors without a recent vaccination history, but it would be beneficial to include a control group with recent smallpox vaccinations for comparison.

Thank you for the comment. We are aware of this limit. Unfortunately, smallpox vaccination or a booster of smallpox vaccination was stopped in Italy in 1981; therefore, we do not have the chance to study this cohort. However, the results are reliable, because the antigen-specific immune response between MPXV and smallpox are similar and the memory T cells induced by the smallpox vaccination are long-lived.

Reviewer 3 Report

Comments and Suggestions for Authors

The authors have investigated CD4 and CD8 T cell specific responses to monkeypox in 16 patients with cured monkeypox and with controls.  They demonstrated mixed Th1 and Th2 responses.  The study is informative, although the number of patients is on the low side.  A few comments are listed below for overall improvement.

1.     Methods, section. 2.6. Provide the name of the company that made the secondary antibodies: Fluorescent tagged antibody to IgG, to IgM and IgA.

2.      Results, section 3.1 and Table 1. IgM response, line 151.  The text states that there were no positive IgM responses.  Table1 lists one positive IgM response.  Explain the discrepancy?

3.     Results, IgA response in Table 1.  

Table 1 lists 7 patients with positive IgA response.  Add this result into the written text.  Was the one positive IgM patient included among the 7 positive IgA patients?  Provide answer in the text.

4.     Discussion.  Re-write paragraph from lines 248-250 about antibody.  Add comments about IgM and IgA responses after monkeypox.  Refer back to data in Table 1. How long after the acute monkeypox illness are IgM and IgA responses detectable?  Provide a reference.

5.     New article for Discussion.  There is a new article published by G. B. da Silva et al, Cytokine storm in human monkeypox; CYTOKINE, 5 March 2024.  (PMID: 38447385). Please add a few comments in the Discussion to explain whether your current data or older data that you cited provide evidence for a cytokine storm in severe cases of monkeypox.

6.     Add a new reference about duration of IgM and IgA titers after monkeypox.  Add the article from CYTOKINE.

Author Response

REVIEWER 3

Open Review

Comments and Suggestions for Authors

The authors have investigated CD4 and CD8 T cell specific responses to monkeypox in 16 patients with cured monkeypox and with controls.  They demonstrated mixed Th1 and Th2 responses.  The study is informative, although the number of patients is on the low side.  A few comments are listed below for overall improvement.

  1. Methods, section. 2.6. Provide the name of the company that made the secondary antibodies: Fluorescent tagged antibody to IgG, to IgM and IgA.

Thank you for the comment, we added the information in the text.

  1. Results, section 3.1 and Table 1. IgM response, line 151.  The text states that there were no positive IgM responses.  Table1 lists one positive IgM response.  Explain the discrepancy?

Thank you for the comment. We meant that most of the Mpox-infected individuals showed a positive IgG and Neutralizing Antibody response and no IgM response. We explained it in the text according to your suggestion (section: Demographic and clinical characteristics of the study population)

  1. Results, IgA response in Table 1.  

Table 1 lists 7 patients with positive IgA response.  Add this result into the written text.  Was the one positive IgM patient included among the 7 positive IgA patients?  Provide answer in the text.

Thank you for the comment. We explained it in the text according to your suggestion (section: Demographic and clinical characteristics of the study population)

  1. Discussion.  Re-write paragraph from lines 248-250 about antibody.  Add comments about IgM and IgA responses after monkeypox.  Refer back to data in Table 1. How long after the acute monkeypox illness are IgM and IgA responses detectable?  Provide a reference.

Thank you for your comment. The serology response of Mpox-cured individuals was characterized by the presence of IgG and Neutralizing antibodies abs (abs) and the absence of IgM response except for one subject. Moreover, 43 % of the cured Mpox subjects still showed a positive IgA response. These results are supported by recent findings from our Institution showing that in Mpox-infected individuals after 6-8 months from symptoms onset, the IgM response became undetectable; the IgA response was detected in 25% of subjects and IgG and Neutralizing abs  decreased but were still positive in all individuals. We evaluated the Ig titers performing an IFA on homemade slides containing MPXV-infected VeroE6 cells and this procedure likely led to a more sensitive assay compared to a previous study reporting an important decline of both IgG and IgA responses after 6 months (Moraes-Cardoso Lancet 2024). These comments have been added to the discussion.

Silva et al, Cytokine storm in human monkeypox; CYTOKINE, 5 March 2024.  (PMID: 38447385). Please add a few comments in the Discussion to explain whether your current data or older data that you cited provide evidence for a cytokine storm in severe cases of monkeypox.

Thank you for your comment, we updated the text as requested.

  1. Add a new reference about duration of IgM and IgA titers after monkeypox.  Add the article from CYTOKINE.

Thank you for your comment, we updated the text as requested.